# CONNECTIVITY MATTERS: NEURAL NETWORK PRUNING THROUGH THE LENS OF EFFECTIVE SPARSITY

## ABSTRACT

Neural network pruning is a fruitful area of research with surging interest in high sparsity regimes. Benchmarking in this domain heavily relies on faithful representation of the sparsity of subnetworks, which has been traditionally computed as the fraction of removed connections (direct sparsity). This definition, however, fails to recognize unpruned parameters that detached from input or output layers of underlying subnetworks, potentially underestimating actual effective sparsity: the fraction of inactivated connections. While this effect might be negligible for moderately pruned networks (up to $10\times$–$100\times$ compression rates), we find that it plays an increasing role for thinner subnetworks, greatly distorting comparison between different pruning algorithms. For example, we show that effective compression of a randomly pruned LeNet-300-100 can be orders of magnitude larger than its direct counterpart, while no discrepancy is ever observed when using SynFlow for pruning (Tanaka et al., 2020). In this work, we adopt the lens of effective sparsity to reevaluate several recent pruning algorithms on common benchmark architectures (e.g., LeNet-300-100, VGG-19, ResNet-18) and discover that their absolute and relative performance changes dramatically in this new, and as we argue, more appropriate framework. To aim for effective, rather than direct, sparsity, we develop a low-cost extension to most pruning algorithms. Further, equipped with effective sparsity as a reference frame, we partially reconfirm that random pruning with appropriate sparsity allocation across layers performs as well or better than more sophisticated algorithms for pruning at initialization (Su et al., 2020). In response to this observation, using a simple analogy of pressure distribution in coupled cylinders from thermodynamics, we design novel layerwise sparsity quotas that outperform all existing baselines in the context of random pruning.

## 1 INTRODUCTION

Recent successful advances of Deep Neural Networks are commonly attributed to their high architectural complexity and excessive size (*over-parametrization*) (Denton et al., 2014; Neyshabur et al., 2019; Arora et al., 2018). Modern state-of-the-art architectures exhibit enormous parameter overhead, requiring prohibitive amounts of resources during both training and inference and leaving a significant environmental footprint (Shoeybi et al., 2019). In response to these challenges, much attention has turned to compression of neural networks and, in particular, parameter pruning. While initial approaches mostly focused on pruning models after training (LeCun et al., 1990; Hassibi et al., 1993), contemporary algorithms optimize the sparsity structure of a network while training its parameters (Mocanu et al., 2018; Evci et al., 2020) or even remove connections before any training whatsoever (Lee et al., 2019; Wang et al., 2020).

Compression rates usually considered in the pruning literature usually fall between $10\times$ and $100\times$ of the size of the original model. However, as contemporary model sizes grow into the billions of parameters, studying higher compression regimes becomes increasingly important. Recently, a new bold sparsity benchmark was set by Tanaka et al. (2020) with Iterative Synaptic Flow (SynFlow), a data-agnostic algorithm for pruning at initialization. Reportedly, it is capable of removing all but only a few hundreds of parameters (a $100,000\times$ compression for VGG-16) and still produce trainable subnetworks, while other pruning methods disconnect networks at much lower sparsity levels (Tanaka et al., 2020). Related work by de Jorge et al. (2021) proposes an iterative version of

one-shot pruning algorithm, Single-shot Network Pruning (SNIP) (Lee et al., 2019), and evaluates it in a similar high sparsity regime, reaching more than $10,000\times$ compression ratio.

**Effective sparsity.** This increased focus on extreme sparsity leads us to consider *what sparsity is meant to represent* in neural networks and computational graphs at large. In the context of neural network pruning, sparsity to date is computed straightforwardly as the *fraction of removed connections* (*direct sparsity*)—and compression as the inverse fraction of unpruned connections (*direct compression*). We observe that this definition does not distinguish between connections that have actually been pruned, and those that have become *effectively* pruned because they have disconnected from the computational flow. In this work, we propose to instead focus on *effective sparsity*—the *fraction of inactivated connections*, be it through direct pruning or through otherwise disconnecting from either input or output of a network (see Figure 1 for an illustration).

In this work, we advocate that effective sparsity (effective compression) be used universally in place of its direct counterpart since it more accurately depicts what one would reasonably consider the network's sparsity state. Using the lens of effective compression for benchmarking allows for a fairer comparison between different unstructured pruning algorithms. Note that effective compression is lower bounded by direct compression, which means that some pruning algorithms will give improved sparsity-accuracy trade-offs in this new framework. In Section 3, we critically reexamine a plethora of recent pruning algorithms for a variety of architectures to find that, in this refined framework, conclusions drawn in previous works appear overstated or incorrect. Figure 2 gives a sneak-preview of this effect for three ab-initio pruning algorithms: SynFlow (Tanaka et al., 2020), SNIP (Lee et al., 2019) and plain random pruning for LeNet-300-100 on MNIST. While SynFlow appears superior to other methods when evaluated against direct compression, it loses its advantage in the effective framework. Such radical performance changes are partly explained by differing gaps between effective and direct compression inherent to different pruning algorithms (Figure 2). We can see that significant departure between direct and effective compression kicks in at relatively low rates below $100\times$, making our work relevant even in these moderate regimes. For example, using random pruning to compress LeNet-300-100 by $100\times$ (sparsity 99%) results in $\sim 1,000\times$ effective compression; yet, removing the same number of parameters with SynFlow yields an unchanged $100\times$ effective compression. What makes certain iterative algorithms like SynFlow less likely to amass disconnected edges? In Section 3, we show that they are fortuitously designed to achieve a close con-

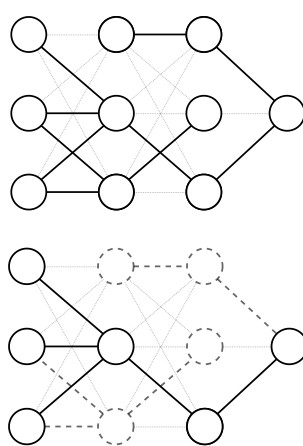

Figure 1: Pruning 11 edges from a fully-connected 21-edge network. Top: direct sparsity $(11/21)$ does not account for disconnected edges (compression $21/10 = 2.1$). Bottom: effective sparsity $(16/21)$ accounts for the 5 dashed connections incident to inactivated neurons (yielding twice as large effective compression $21/5 = 4.2$).

vergence of direct and effective sparsity, hinting that preserving connectivity is an important aspect in the strong performance of high-compression pruning algorithms (Tanaka et al., 2020; de Jorge et al., 2021). Moreover, the lens of effective compression gives access to more extreme compression regimes for some pruning algorithms, which appear to disconnect much earlier when not accounting for inactive connections. For these high effective compression ratios all three pruning methods from Figure 2 perform surprisingly similar, even though they use varying degrees of information on data and parameter values.

**Layerwise Sparsity Quotas (LSQ) and Ideal Gas Quotas (IGQ).** A recent thread of research by Frankle et al. (2021) and Su et al. (2020) shows that performance of trained subnetworks produced by algorithms for pruning at initialization is robust to randomly reshuffling unpruned edges within layers before training. This observation led to the conjecture that these algorithms essentially generate successful distributions of sparsity across layers, while the exact connectivity patterns are unimportant. In Section 4, we reexamine this conjecture through the lens of effective sparsity, confirm it for moderate compression regimes $(10\times-100\times)$ studied by Frankle et al. (2021) and Su et al. (2020), but find the truth to be more nuanced at higher compression rates. Nonetheless, this re-

sult highlights the importance of algorithms that carefully engineer *layerwise sparsity quotas (LSQ)* to obtain very simple and adequately performing pruning algorithms that are data- and parameter-agnostic. Another important motivation to search for good LSQ is that global pruning algorithms frequently remove entire layers prematurely (Lee et al., 2020) (cf. layer-collapse (Tanaka et al., 2020)), even before any significant differences between direct and effective sparsity emerge. Well-engineered LSQ could avoid this and enforce proper redistribution of compression across layers (see (Gale et al., 2019; Mocanu et al., 2018; Evci et al., 2020) for existing baselines). In Section 4, we propose a novel LSQ coined *Ideal Gas Quotas (IGQ)* by drawing intuitive analogies from physics. Effortlessly computable for any network-sparsity combination, IGQ performs similarly or better than any other baseline in the context of random pruning at initialization and of magnitude pruning after training.

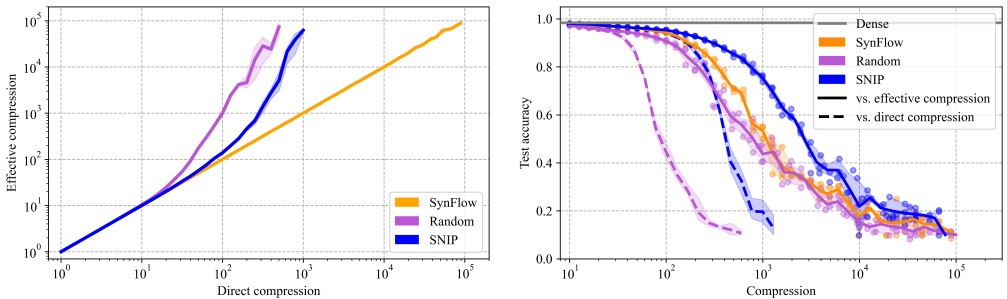

Figure 2: LeNet-300-100 trained on MNIST after pruning with SNIP, SynFlow and layerwise uniform random pruning. Left: gaps between direct and effective compression. Right: SynFlow has a better sparsity-accuracy trade-off than SNIP when plotted against direct compression (dashed), but not against effective compression (solid curves fitted to dots that represent inidvidual experiments). Dashed and solid curves coincide for SynFlow.

**Effective pruning.** Pruning to any desired direct sparsity is straightforward: one simply needs to mask out the corresponding number of parameters from a network. Effective sparsity, unfortunately, is more unpredictable and difficult to control. In particular, several known pruning algorithms suffer from layer-collapse once reaching a certain sparsity level, leading to unstable effective sparsity just before the disconnection. As a result, most pruning methods are unable to deliver certain values of effective sparsity regardless of how many connections are pruned. When possible, however, one needs to carefully tune the number of pruned parameters so that effective sparsity lands near a desired value. In Section 5, we suggest a simple extension to algorithms for pruning at initialization or after training that helps bring effective sparsity close to any predefined achievable value while incurring costs that are at most logarithmic in model size.

**Our contributions.** In this study, we ($i$) clearly formulate and illustrate the importance of effective sparsity by reevaluating several recent pruning strategies; ($ii$) provide algorithms to prune according to and compute effective sparsity; ($iii$) reconfirm the hypothesis from (Frankle et al., 2021) in the new framework and design efficient layerwise sparsity quotas IGQ that perform consistently well across all sparsity regimes.

## 2 RELATED WORK

Neural network compression encompasses a number of orthogonal approaches such as parameter regularization (Lebedev & Lempitsky, 2016; Louizos et al., 2018), variational dropout (Molchanov et al., 2017), vector quantization and parameter sharing (Gong et al., 2014; Chen et al., 2015; Han et al., 2016), low-rank matrix decomposition (Denton et al., 2014; Jaderberg et al., 2014), and knowledge distillation (Buciluǎ et al., 2006; Hinton et al., 2015). Network pruning, however, is by far the most common technique for model compression, and can be partitioned into structured (at the level of entire neurons/units) and unstructured (at the level of individual connections). While the former offers resource efficiency unconditioned on use of specialized hardware (Liu et al., 2019) and constitutes a fruitful research area (Li et al., 2017; Liu et al., 2017), we focus on the more actively studied

unstructured pruning, which is where differences between effective and direct sparsity emerge. In what follows we give a quick overview, naturally grouping pruning methods by the time they are applied relative to training (see (Frankle & Carbin, 2019) and (Wang et al., 2020) for a similar taxonomy).

**Pruning after training.** These earliest pruning techniques were designed to remove the least "salient" learned connections without sacrificing predictive performance. Optimal Brain Damage (LeCun et al., 1990) and its sequel Optimal Brain Surgeon (Hassibi et al., 1993) use the Hessian of the loss to estimate sensitivity to removal of individual parameters. Han et al. (2015) popularized magnitude as a simple and effective pruning criterion. It proved to be especially successful when applied alternately with several finetuning cycles, which is commonly referred to as Iterative Magnitude Pruning (IMP), a modification of which was used by Frankle & Carbin (2019) to discover lottery tickets. Later, Dong et al. (2017) showed that magnitude-based pruning minimizes $\ell_2$ distortion of each layer's output incurred by parameter removal. Recently, Lee et al. (2021) extend this idea and propose Layer-Adaptive Magnitude-Based Pruning (LAMP), which approximately minimizes the upper bound of the $\ell_2$ distortion of the entire network. While equivalent to magnitude pruning within individual layers, LAMP automatically discovers state-of-the-art layerwise sparsity quotas (see Section 4) that yield better performance (as a function of *direct* compression) than existing alternatives in the context of IMP.

**Pruning during training.** Algorithms in this category learn sparsity structures together with parameter values, hoping that continued training will correct for damage incurred by pruning. To avoid inefficient prune-retrain cycles inherent to IMP, Narang et al. (2017) introduce gradual magnitude pruning over a single training round. Subsequently, Zhu & Gupta (2018) modify this algorithm by introducing a simpler pruning schedule and keeping layerwise sparsities uniform throughout training. Sparse Evolutionary Training (SET) (Mocanu et al., 2018) starts with an already sparse subnetwork and restructures it during training by pruning and randomly reviving connections. Unlike SET, Mostafa & Wang (2019) allow redistribution of sparsity across layers, while Dettmers & Zettlemoyer (2019) use gradient momentum as the criterion for parameter regrowth. Evci et al. (2020) rely on the instantaneous gradient to revive weights but follow SET to maintain the initial layerwise sparsity distribution during training. A different body of works tackle the general optimization problem with an intractable $\ell_0$ parameter sparsity constraint by designing and solving related continuous problems (Zhou et al., 2021; Savarese et al., 2020; Kusupati et al., 2020). For example, Continuous Sparsification (CS) by Savarese et al. (2020) uses a sigmoid of learnable continuous variables as mask values and applies $\ell_1$ regularization, effectively forcing them to either 0 or 1 in training.

**Pruning before training.** Pruning at initialization is especially alluring to deep learning practitioners as it promises lower costs of both optimization and inference. While this may seem too ambitious, the Lottery Ticket Hypothesis (LTH) postulates that randomly initialized dense networks do indeed contain highly trainable and equally well-performing sparse subnetworks (Frankle & Carbin, 2019). Inspired by the LTH, Lee et al. (2019) design SNIP, which uses connection sensitivity as a parameter saliency score. Wang et al. (2020) notice that SNIP creates bottlenecks or even removes entire layers and propose Gradient Signal Preservation (GraSP) as an alternative that aims to maximize gradient flow in a pruned network. de Jorge et al. (2021) improve SNIP by applying it iteratively, allowing for reassessment of saliency scores during pruning and helping networks stay connected at higher compression rates. A truly new compression benchmark was set by Tanaka et al. (2020); their algorithm, SynFlow, iteratively prunes subsets of parameters according to their $\ell_1$ path norm and helps networks reach maximum compression without disconnecting. For example, SynFlow achieves non-random test accuracy on CIFAR-10 with a $100,000\times$ compressed VGG-16, while SNIP and GraSP fail already at $100\times$ and $1,000\times$, respectively. An extensive ablation study by Frankle et al. (2021) examines SNIP, GraSP and SynFlow within moderate compression rates (up to $100\times$) and reveals that performance of subnetworks produced by these methods is stable under layerwise rearrange prior to training. Later, this result was independently confirmed by Su et al. (2020) for SNIP and GraSP only. This observation suggests that these algorithms perform as well as random pruning with corresponding layerwise quotas, putting the spotlight on designing competitive LSQ (Mocanu et al., 2018; Gale et al., 2019; Lee et al., 2021).

## 3 EFFECTIVE SPARSITY

In this section, we present our comparisons of a variety of pruning algorithms under the lens of effective compression. To illustrate the striking difference between direct and effective sparsity and expose the often radical change in relative performance of pruning algorithms when switching from the former to the latter, we evaluate several recent methods (SNIP, GraSP, SynFlow, LAMP[1], CS[2], and SNIP-iterative) and random pruning with uniform sparsity distribution across layers in both frameworks. Our experiments encompass modern architectures on commonly used computer vision benchmark datasets: LeNet-300-100 (Lecun et al., 1998) on MNIST, LeNet-5 (Lecun et al., 1998) on CIFAR-10, VGG-19 (Simonyan & Zisserman, 2015) on CIFAR-100 , and ResNet-18 (He et al., 2016) on TinyImageNet. We place results of VGG-16 (Simonyan & Zisserman, 2015) on CIFAR-10 in Appendix B, as they closely resemble those of VGG-19. Further experimental details are presented in Appendix A.

**Notation.** Consider an $L$-layer neural network $f(\boldsymbol{\Theta}; x)$ with weight tensors $\boldsymbol{\Theta} = \{\Theta_\ell\}_{\ell=1}^L$ for $\ell \in [L]$. A subnetwork is specified by a set of binary masks that indicate unpruned parameters $M_\ell \in \{0,1\}^{|\Theta_\ell|}$. With $\mathbf{M} = \{M_\ell\}_{\ell=1}^L$, it is given by $f(\boldsymbol{\Theta} \odot \mathbf{M}; x)$ where $\odot$ indicates pointwise multiplication. Note that biases and batch normalization parameters (Ioffe & Szegedy, 2015) are normally considered unprunable. Direct sparsity, the fraction of pruned weights, is given by $s(\mathbf{M}) = 1 - \sum_\ell \|M_\ell\|_0 / \sum_\ell |M_\ell|$ and direct compression rate is defined as $(1 - s(\mathbf{M}))^{-1}$.

Figure 3 reveals that different algorithms tend to develop varying amounts of inactive connections. For example, effective compression of subnetworks pruned by LAMP consistently reaches $10\times$ of their direct compression across all architectures, at which point at least nine in ten unpruned connections are effectively inactivated. Other methods (e.g., SNIP on VGG-19) remove entire layers early on, before any substantial differences between effective and direct compression emerge. SNIP-iterative and especially SynFlow, however, demonstrate a truly unique property: subnetworks pruned by these two algorithms exhibit practically equal effective and direct compressions, and, in the case of SynFlow, disconnect only at very high compression rates. What makes them special? Both SynFlow and SNIP-iterative are multi-shot pruning algorithms that remove parameters over 100 and 300 iterations, respectively. SynFlow ranks connections by their $\ell_1$ path norm (sum of weighted paths passing through the edge, where the weight of a path is the product of magnitudes of weights of its edges). SNIP uses connection sensitivity scores from Lee et al. (2019) $\left|\frac{\partial \mathcal{L}}{\partial \theta}\theta\right|$ as a saliency measure, where $\mathcal{L}$ is the loss function. Both these pruning criteria assign the lowest possible score of zero to inactive connections, scheduling them for immediate removal in the subsequent pruning iteration. Thus, by virtue of their iterative design, these two methods produce subnetworks with little to no difference between effective and direct compression. They are fortuitously designed to prune inactivated edges, which might explain their high-compression performance.

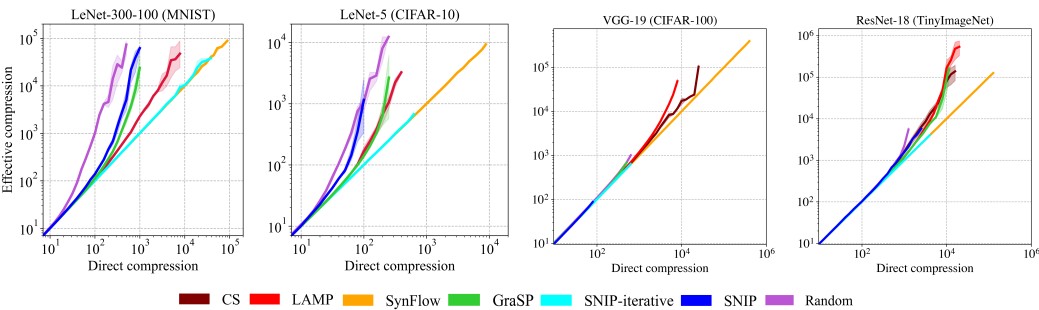

Figure 3: Effective versus direct compression across different pruning methods and architectures (curves and bands represent min/average/max across 3 seeds where subnetworks disconnect last among a total of 5 seeds).

---

[1] as a state-of-the-art representative of magnitude pruning after training and, in particular, lottery tickets (Frankle & Carbin, 2019).

[2] as a representative of methods concerned with learnable sparsity

Tanaka et al. (2020) compare SynFlow to SNIP and GraSP using direct sparsity, claiming it vastly superior in high compression regimes. However, pruning methods that generate large amounts of inactivated connections are clearly at a significant disadvantage in the original direct framework. Figure 4 shows that the performance gap between SynFlow and other methods shrinks on all tested architectures under effective compression. The most dramatic changes are perhaps evident with LeNet-300-100 where SynFlow significantly dominates both SNIP and GraSP in direct comparison, but becomes strictly inferior when taken to the more telling effective compression. On the other hand, differences are not as pronounced on purely convolutional architectures such as VGG-19, and ResNet-18. Feature maps in convolutional layers are connected via groups of several parameters (kernels), making them more robust to inactivation compared to neurons in fully-connected layers.

*Computing effective sparsity:* In advocating the use of effective sparsity, we must make sure that it can be calculated efficiently. We propose an easily computable approach leveraging SynFlow. Note that a connection is inactive if and only if it is not part of any path from input to output. Assuming that unpruned weights are non-zero, this is equivalent to having zero $\ell_1$ path norm. Tanaka et al. (2020) observe that path norms can be efficiently computed with one pass on the all-ones input as $\left|\frac{\partial \mathcal{R}}{\partial \theta} \theta\right|$, where $\mathcal{R} = \mathbb{1}^\top f'(|\Theta| \odot \mathbf{M}, \mathbb{1})$ and $f'$ is the linearized version of the original network $f$. For deep models, rescaling of weights is required to avoid numerical instability (Tanaka et al., 2020).

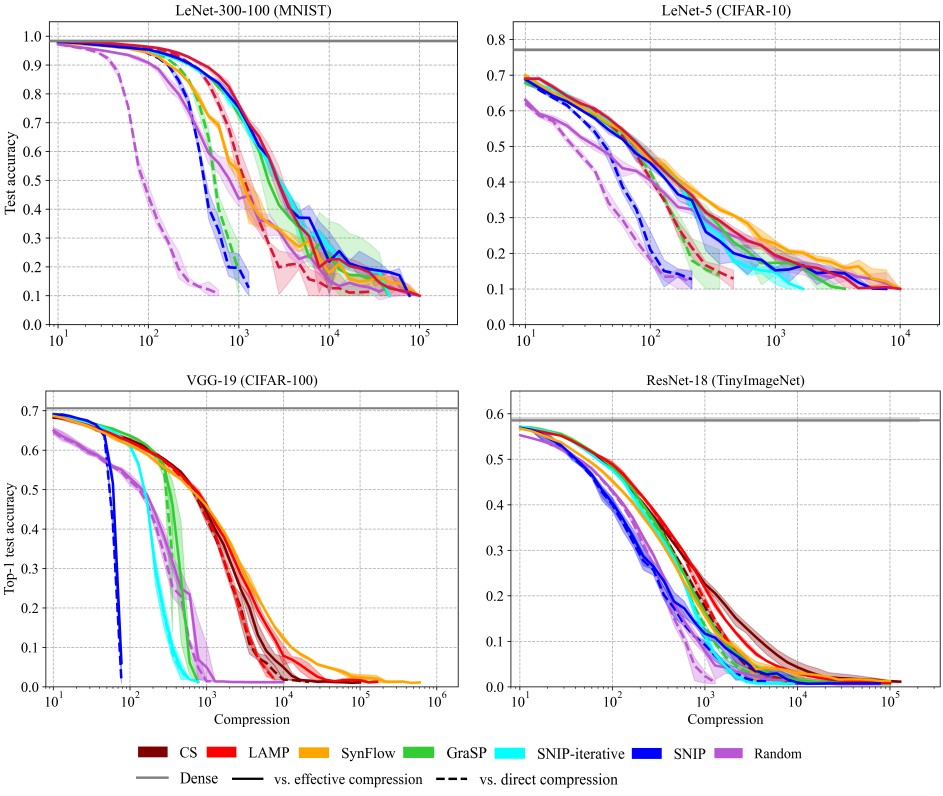

Figure 4: Test accuracy (min/average/max) of subnetworks trained from scratch after being pruned by different algorithms plotted against direct (dashed) and effective (solid) compression. Dashed and solid curves overlap for SynFlow and SNIP-iterative. Solid curves are fitted to scatter data (not shown for clarity of the presentation) as in Figure 2.

# 4 LAYERWISE SPARSITY QUOTAS (LSQ) AND A NOVEL ALLOCATION METHOD (IGQ)

Inspired by Frankle et al. (2021) and Su et al. (2020), we wish to confirm that SNIP, GraSP, and SynFlow work no better than random pruning with corresponding layerwise sparsity allocation.

While Frankle et al. (2021) and Su et al. (2020) only considered moderate compression rates up to $100\times$ and used direct sparsity as a reference frame, we reconfirm their conjecture in the effective framework and test it across the entire compression spectrum. We generate and train two sets of subnetworks: $(i)$ pruned by either SNIP, GraSP, and SynFlow (*original*), and $(ii)$ randomly pruned while preserving layerwise sparsity quotas provided by each of these three methods (*random*).

Our results in Figure 5 agree with observations made by Frankle et al. (2021) and Su et al. (2020): in the $10\times$–$100\times$ compression range, all three random pruning algorithms perform similarly (LeNet-300-100, VGG-19) or better (ResNet-18) than their original counterparts. Effective sparsity allows us to faithfully examine higher compression, where the evidence is more equivocal. Similar patterns are still seen on ResNet-18; however, the original SNIP and GraSP beat random pruning with corresponding layerwise sparsities by a wide margin starting at $100\times$ compression on LeNet-300-100. Random pruning associated with SynFlow matches original SynFlow on the same network for longer, up to $1,000\times$ compression. On VGG-19, SynFlow bests the corresponding random pruning from about $500\times$ compression onward, while the original SNIP suffers from disconnection early on together with its random variant. Despite these nuances in the high compression regime, random pruning with specific layerwise sparsity quotas fares extremely well in the moderate sparsity regime (up to $99\%$) and is even competitive to full-fledged SynFlow (see Figure 6). Therefore, random pruning can be a cheap and competitive alternative to more sophisticated and resource-consuming algorithms. Random methods from Figure 5, however, still require running SNIP, GraSP, or SynFlow to identify appropriate sparsity quotas and thus are just as expensive. Furthermore, sparsity distributions inherited from global pruning methods frequently suffer from premature removal of entire layers (e.g., SNIP on VGG-19), which is undesired. Can we engineer readily computable and consistently well-performing sparsity quotas?

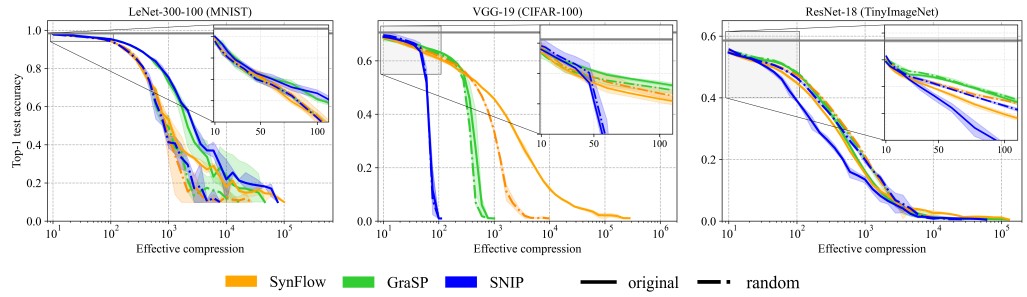

Figure 5: Original methods for pruning at initialization (solid) and random pruning with corresponding layerwise sparsity quotas (dashdot). Test accuracy of the unpruned network is shown in grey.

To our knowledge, there are only a few *ab-initio* approaches in the literature to allocate sparsity in a principled fashion. *Uniform* is the simplest solution that keeps sparsity constant across all layers. Gale et al. (2019) give a modification (denoted *Uniform+* following Lee et al. (2021)) that retains all parameters in the first convolutional layer and caps sparsity of the last fully-connected layer at $80\%$. A more sophisticated approach, *Erdös-Renyi-Kernel (ERK)*, sets the density of a convolutional layer with kernel size $w \times h$, fan-in $n_{in}$ and fan-out $n_{out}$ proportional to $(w+h+n_{in}+n_{out})/(w \cdot h \cdot n_{in} \cdot n_{out})$. Although originally used as a sparsity distribution schema for methods with dynamic sparse structres (SET by (Mocanu et al., 2018) and RigL by Evci et al. (2020)), we follow (Lee et al., 2021) and use ERK as a baseline sparsity distribution for sparse-to-sparse training with a fixed subnetwork topology. The last two approaches are unable to support the entire range of sparsities: Uniform+ can only achieve moderate *direct* compression because of the prunability constraints on its first and last layer, while both direct and effective sparsity levels achievable with ERK are often lower bounded. For example, the density of certain layers of VGG-16 set by ERK exceeds 1 when cutting less than $99\%$ of parameters, unless excessive density is redistributed. We suggest a formal definition for layerwise sparsity quotas to guide future research into sparsity allocation and avoid problems that riddle Uniform+ and ERK.

**Definition 1 (Layerwise Sparsity Quotas).** *A function $\mathcal{Q}\colon [0,1] \to [0,1]^L$ mapping a target sparsity $s$ to layerwise sparsities $\{s_\ell\}_{\ell=1}^L$ is called Layerwise Sparsity Quotas (LSQ) if it satisfies*

*the following properties: (i) total sparsity: for any $s \in [0,1]$, $s \sum_\ell |\Theta_\ell| = \sum_\ell s_\ell |\Theta_\ell|$, and (ii) monotonicity: $[\mathcal{Q}(s_1)]_\ell \leq [\mathcal{Q}(s_2)]_\ell$ for any $\ell \in [L]$ whenever $s_1 \leq s_2$.*

We now present *Ideal Gas Quotas (IGQ)*, our formula for sparsity allocation that satisfies Definition 1 and outperforms the above mentioned baselines, while faring very well (over effective sparsity) compared to the allocation quotas derived from sophisticated pruning methods such as SynFlow. To develop an intuition on what constitutes a good LSQ construction, we study the layerwise sparsities induced by contemporary pruning algorithms such as LAMP and SynFlow. As a rule, they $(i)$ prune larger layers more aggressively than smaller layers, i.e., $[\mathcal{Q}(s)]_\ell \leq [\mathcal{Q}(s)]_{\ell'}$ whenever $|\Theta_\ell| \leq |\Theta_{\ell'}|$ for any $s \in [0,1]$, and $(ii)$ avoid premature removal of entire layers, i.e., $[\mathcal{Q}(s)]_\ell = 1$ if and only if $s = 1$ (see Appendix C for illustration).

Our approach is to interpret compression of layers within a network as compression of stacked cylinders of unit volume filled with gas, where the height of the cylinder is proportional to the number of parameters in that layer. We then use the Ideal Gas Law to derive the compression of each of the stacked coupled cylinders. More formally, model each layer $\ell \in [L]$ as a cylinder of height $H_\ell = |\Theta_\ell|$ and cross-section area $S_\ell = |\Theta_\ell|^{-1}$. Further, assume that these stacked weightless cylinders with frictionless pistons and filled with the same amount of ideal gas $\nu$ are in thermodynamic equilibrium with common pressure $P_0 = 1$ and temperature $T$. Isothermal compression of this system using an external force $F$ is governed by the Ideal Gas Law: $P'_\ell V'_\ell = \nu RT = P_0 V_0$, where $P'_\ell = F/S_\ell + P_0$, $V'_\ell = S_\ell h_\ell$, and $h_\ell$ is its new compressed height. Then, $P_0 H_\ell S_\ell = (F/S_\ell + P_0)S_\ell h_\ell$ or, equivalently, $H_\ell/h_\ell = F/(P_0 S_\ell) + 1 = F|\Theta_\ell| + 1$. Interpreting $H_\ell/h_\ell$ as compression ratio of layer $\ell$, we arrive at compression quotas $\{F|\Theta_\ell|+1\}_{\ell=1}^L$ (or sparsity quotas $\{1 - (F|\Theta_\ell| + 1)^{-1}\}_{\ell=1}^L$) parameterized by the force $F$ controlling the overall sparsity of the network. Given a target sparsity $s \in [0,1]$, the needed value of $F$ can simply be found with binary search to any desired precision. Our IGQ clearly satisfies all conditions of Definition 1 and the other properties identified above.

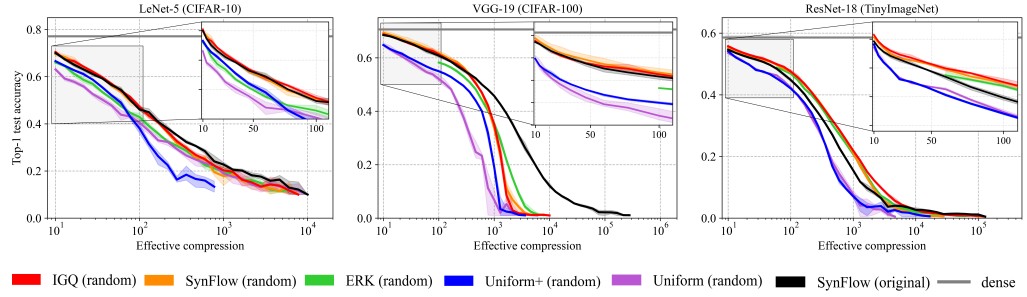

Figure 6: Test performance of trained subnetworks after random pruning with different layerwise sparsity distributions. Original SynFlow (black) is shown for reference.

We now evaluate IGQ for random pruning, comparing it against ERK, Uniform, Uniform+, as well as random pruning with the sparsity quotas induced by SynFlow for reference (Figure 6). Across all architectures, random pruning with IGQ and SynFlow sparsity quotas are almost indistinguishable from each other, suggesting that IGQ successfully mimics the quotas produced by SynFlow, which requires substantial effort to compute. While ERK sometimes exhibits similar (ResNet-18) or even better (VGG-19 compressed to $1,000\times$ or higher) performance than IGQ, it yields invalid layerwise sparsity quotas when removing less than $98\%$ and $99\%$ of parameters from ResNet-18 and VGG-19, respectively, thus failing to satisfy Definition 1. In the moderate sparsity regime (up to $99\%$), subnetworks pruned by IGQ reach unparalleled performance after training. Therefore, judging by a tripartite criterion of test performance, compliance with Definition 1, and computational efficiency, IGQ beats all considered baselines.

In Appendix C, we test IGQ in the context of magnitude pruning after training. Here, performance of IGQ practically coincides with that of LAMP, making it the only known LSQ to consistently perform best and a competitive method for layerwise sparsity allocation.

## 5 EFFECTIVE PRUNING

Unlike pruning to a target direct sparsity, pruning to achieve a particular *effective* sparsity can be non-trivial. Here, we present an extension to algorithms for pruning at initialization or after training that achieves this goal efficiently, when possible (see Figure 7).

**Ranking-based pruning.** Algorithms like GraSP, Syn-Flow, and LAMP rank parameters by some notion of importance to guide pruning. When such a ranking $R\colon \Theta \to \mathbb{R}$ is available, the naive solution is to iterate through all scores in order, considering each as a potential pruning threshold $t$ and recording effective sparsity of the corresponding subnetwork with parameters $\{\theta \in \Theta\colon R[\theta] < t\}$ removed. While provably identifying the optimal threshold that yields a subnetwork with effective sparsity as close to the desired value as possible, this approach requires $\mathcal{O}(|\Theta|)$ prune-evaluate cycles, which is unreasonable for most contemporary architectures. To achieve an efficient overhead of $\mathcal{O}(\log|\Theta|)$ at the price of minor inaccuracy, we utilize binary search for the cut-off threshold instead, leveraging the following monotonicity property: given two pruning thresholds $t_1, t_2 \in R$ and corresponding subnetworks $S_1, S_2$, we have $t_1 \leq t_2$ if and only if $S_2 \subseteq S_1$, which implies EffectiveSparsity$(S_1) \leq$ EffectiveSparsity$(S_2)$ (note that in general Sparsity$(S_1) \leq$ Sparsity$(S_2)$ does not imply the last inequality above). Thus, binary search will branch in the correct direction. In Appendix D, inspired by the performance of random pruning as presented in Section 4, we describe a modification of this algorithm to allow for effective random pruning where parameter rankings are not available.

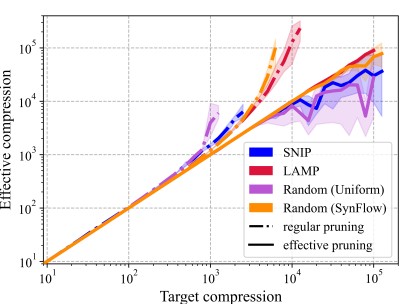

Figure 7: Effective compression produced by regular (dashdot) and our effective (solid) pruning on ResNet-18 according to ranking-based (left) and random (right) algorithms. Our procedures help pruning reach target effective sparsity, falling short only when the subnetwork is on the brink of disconnection.

## 6 DISCUSSION

In our work, we argue that *effective sparsity (effective compression)* is the correct benchmarking measure for pruning algorithms since it discards effectively inactive connections and represents the true remaining connectivity pattern. Moreover, effective sparsity allows us to study extreme compression regimes for subnetworks that otherwise appear disconnected at much lower direct sparsities. We initiate the study of current pruning algorithms in this refined frame of reference and rectify previous benchmarks. To facilitate the use of effective sparsity in future research, we describe low-cost procedures to both compute and achieve desired effective sparsity when pruning. Lastly, with effective sparsity allowing us to fairly zoom into higher compression regimes than previously possible, we examine random pruning with prescribed layerwise sparsities and propose our own readily computable quotas (IGQ) after establishing conditions reasonable LSQ should fulfill. We show that IGQ, while allowing for any level of sparsity, is more advantageous than all existing similar baselines (Uniform, ERK) and gives comparable performance to sparsity quotas derived from more sophisticated and computationally expensive algorithms like SynFlow.

*Limitations and Broader Impacts:* We hope that the lens of effective compression will spur more research in high compression regimes. One possible limitation is that it is harder to control effective compression exactly. In particular using different seeds might lead to slightly different effective compression rates. However, these perturbations are minor. Another small limitation is that our effective pruning strategies are not immediately applicable to some algorithms that prune while training (e.g., RigL (Evci et al., 2020)). However, in most cases our approach can be adapted. Lastly, one might argue that for some architectures accuracy drops precipitously with higher compression thus making very sparse subnetworks less practical. We hope that opening the study of high compressions will allow to explore how to use sparse networks as building blocks, for instance using the power of ensembling. Our framework allows a principled study of this regime.

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

## A    EXPERIMENTAL DETAILS

Our experimental work encompasses five different architecture-dataset combinations: LeNet-300-100 (Lecun et al., 1998) on MNIST (Creative Commons Attribution-Share Alike 3.0 license), LeNet-5 (Lecun et al., 1998) and VGG-16 (Simonyan & Zisserman, 2015) on CIFAR-10 (MIT license), VGG-19 (Simonyan & Zisserman, 2015) on CIFAR-100 (MIT license), and ResNet-18 (He et al., 2016) on TinyImageNet (MIT license). Following Frankle et al. (2021), we do not reinitialize sub-networks after pruning (we revert back to the original initialization when pruning a pretrained model by LAMP). We use our own implementation of all pruning algorithms in TensorFlow except for GraSP, for which we use the original code in PyTorch published by Wang et al. (2020). All runs were repeated 3 times for stability of results. Training was performed on an internal cluster equipped with NVIDIA RTX-8000, NVIDIA V-100, and AMD MI50 GPUs. Hyperparameters and training schedules used in our experiments are adopted from related works and are listed in Table 1. We apply standard augmentations to images during training. In particular, we normalize examples per-channel for all datasets and randomly apply: $(i)$ shifts by at most 4 pixels in any direction and horizontal flips (CIFAR-10, CIFAR-100, and TinyImageNet), or $(ii)$ rotations by up to 4 degrees (MNIST).

Table 1: Summary of experimental work. All architectures include batch normalization layers followed by ReLU activations. Models are initialized using Kaiming normal scheme (fan-avg) and optimized by SGD (momentum 0.9) with a stepwise LR schedule ($10\times$ drop factor applied on specified *drop epochs*). The categorical cross-entropy loss function is used for all models.

| Model | Epochs | Drop epochs | Batch | LR | Decay | Source |
|---|---|---|---|---|---|---|
| LeNet-300-100 | 160 | 41/83/125 | 100 | 0.1 | $5e\text{-}4$ | Lee et al. (2019) |
| LeNet-5 | 307 | 76/153/230 | 128 | 0.1 | $5e\text{-}4$ | Lee et al. (2019) |
| VGG-16 | 160 | 80/120 | 128 | 0.1 | $1e\text{-}4$ | Frankle et al. (2021) |
| VGG-19 | 160 | 80/120 | 128 | 0.1 | $5e\text{-}4$ | Wang et al. (2020) |
| ResNet-18 | 200 | 100/150 | 256 | 0.2 | $1e\text{-}4$ | Frankle et al. (2021) |

## B    EXPERIMENTS WITH VGG-16

In Figure 8, we display the results of our experiments with VGG-16 on CIFAR-10. As we argued in Section 3, higher sparsities are required for purely convolutional architectures (such as VGG-16) to develop inactive connections since feature maps are harder to disconnect. At the same time, several algorithms (SNIP, SNIP-iterative, GraSP) suffer from layer-collapse at modest sparsities (99.9% or less) and, hence, fail to develop significant amounts of inactive parameters. For this reason, as evident from Figures 3, 4, and 8, VGG-16 arguably showcases the least differences between effective and direct compression among all tested architectures.

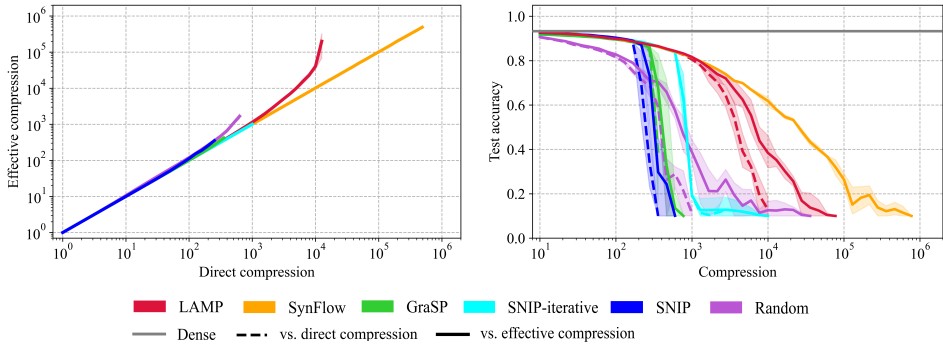

Figure 8: Left: effective versus direct compression of VGG-16 when pruned by different algorithms. Right: test accuracy (min/average/max) of VGG-16 trained from scratch after being pruned by different algorithms plotted against direct (dashed) and effective (solid) compression. Dashed and solid curves overlap for SynFlow and SNIP-iterative.

## C    FURTHER ANALYSIS OF IGQ

To guide the design of new LSQ, we take a closer look at the layerwise compression quotas arrived at by successful global pruning algorithms such as SynFlow and LAMP. As mentioned in Section 4 and as evident from Figure 9, these methods prune larger, parameter-heavy layers more aggressively than smaller layers. Together with the other desired properties discussed in Section 4, this observation partly motivates the design of Ideal Gas Quotas (IGQ), where the number of parameters in the layer corresponds to the height of the cylinder. It is surprising how closely the sparsity quotas achieved by IGQ resemble those of SynFlow considering that they describe a physical process (Figure 9).

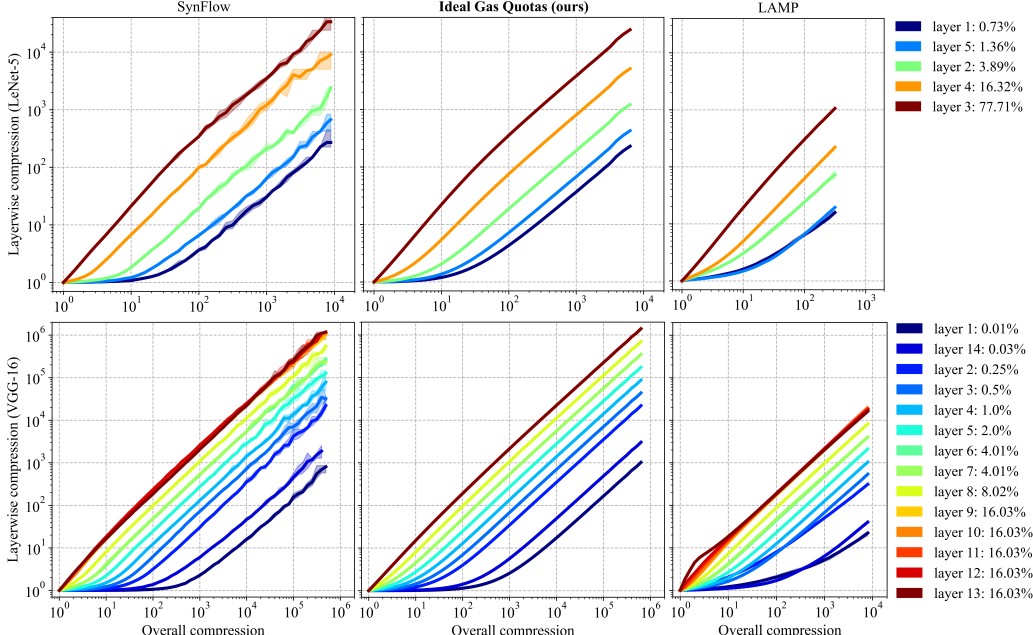

Figure 9: Layerwise direct compression quotas of LeNet-5 (top) and VGG-16 (bottom) associated with SynFlow (left), our IGQ (middle), and LAMP (right). Percentages indicate layer sizes relative to the total number of parameters; colors are assigned accordingly from blue (smaller layers) to red (larger layers). Curves of LAMP and SynFlow end when the underlying network disconnects.

In addition to the ab-initio pruning experiments in Section 4, we test IGQ in the context of magnitude pruning after training. In this set of experiments, we pretrain fully-dense models and prune them by magnitude using global methods (Global Magnitude Pruning, LAMP) or layer-by-layer respecting sparsity allocation quotas (Uniform, Uniform+, ERK, and IGQ). Then, we revert the unpruned weights back to their original random values and fully retrain the resulting subnetworks to convergence. Results are displayed in Figure 10 in the framework of effective compression. Overall, our method for distributing sparsity in the context of magnitude pruning performs consistently well across all architectures and favorably compares to other baselines, especially in moderate compression regimes of $100\times$ or less. Even though Global magnitude pruning can marginally outperform IGQ, it is completely unreliable on VGG-19. ERK appears slighly better than IGQ on VGG-19 and ResNet-18 at extreme sparsities, however, it performs much worse on LeNet-300-100 and has other general deficiencies as we discussed in Section 4. The closest rival of IGQ is LAMP, which performs very similarly but is still unable to reach IGQ's performance on VGG-19 and ResNet-18 in moderate compression regimes. Note, however, that all presented methods require practically equal compute and time; thus, the evidence in Figure 10 is not meant to advertise IGQ as a cheaper alternative to LAMP but rather to illustrate the effectiveness of IGQ.

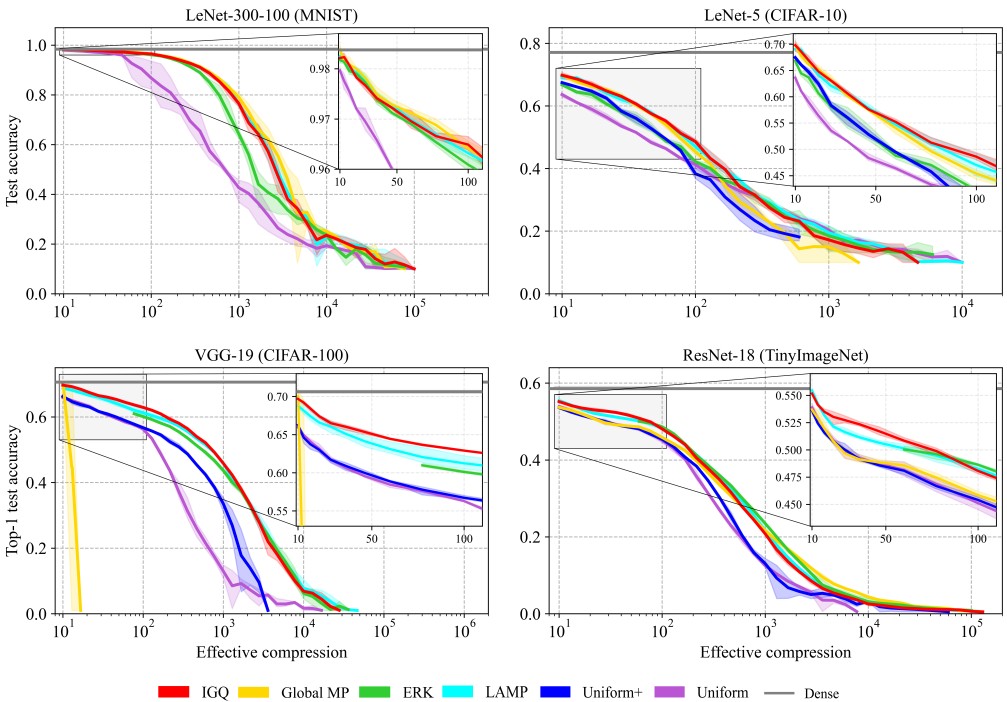

Figure 10: Test performance of retrained subnetworks after pruning with different magnitude-based methods. Uniform+ is not shown for LeNet-300-100 since it is designed for convolutional networks.

## D  EFFECTIVE RANDOM PRUNING

In Section 4, we saw that random pruning with carefully crafted layerwise sparsity quotas $\mathcal{Q}\colon [0,1] \to [0,1]^L$ fares well (especially in the framework of effective sparsity) with more sophisticated pruning methods, proving to be a cheaper and simpler alternative. Effective random pruning is not as straightforward as binary search with ranking-based methods (Section 5) because, for any two subnetworks $S_1$ and $S_2$, $\text{Sparsity}(S_1) \leq \text{Sparsity}(S_2)$ does not imply $\text{EffectiveSparsity}(S_1) \leq \text{EffectiveSparsity}(S_2)$, and random pruning is unlikely to produce a neat chain of embedded subnetworks like ranking-based pruning in Section 5.

---

**Algorithm 1:** Approximate Effective Random Pruning

---

**Input:** Desired effective sparsity $s$; LSQ function $\mathcal{Q}\colon [0,1] \to [0,1]^L$.
$i \leftarrow 0;\ j \leftarrow |\Theta|;\ \mathbf{M}^{(1)} \leftarrow \mathbf{1};\ \mathbf{M}^{(2)} \leftarrow \mathbf{0};\ P_\ell, U_\ell \leftarrow \Theta_\ell$ for all $\ell \in [L]$;
**while** $j - i > 1$ **do**
    $m \leftarrow \lfloor (i+j)/2 \rfloor;\ \{s_\ell\}_{\ell=1}^L \leftarrow \mathcal{Q}(m|\Theta|^{-1})$;
    $T_\ell \leftarrow \text{RandomSelect}(\text{from} = U_\ell \cap P_\ell, \text{size} = s_\ell|\Theta_\ell| - (1 - |U_\ell||\Theta_\ell|^{-1}))$ for all $\ell \in [L]$;
    $M_\ell \leftarrow \text{CreateMask}(\text{pruned} = \Theta_\ell \setminus [U_\ell \setminus T_\ell], \text{unpruned} = U_\ell \setminus T_\ell)$ for all $\ell \in [L]$;
    **if** *EffectiveSparsity*$(\{M_\ell\}) < s$ **then**
        $U_\ell \leftarrow U_\ell \setminus T_\ell$ for all $\ell \in [L];\ \mathbf{M}^{(1)} \leftarrow \{M_\ell\}_{\ell=1}^L;\ i \leftarrow m$;
    **else**
        $P_\ell \leftarrow P_\ell \setminus T_\ell$ for all $\ell \in [L];\ \mathbf{M}^{(2)} \leftarrow \{M_\ell\}_{\ell=1}^L;\ j \leftarrow m$;
    **end**
**end**
**Return:** Masks $\mathbf{M}^{(1)}$ with $\text{EffectiveSparsity}(\mathbf{M}^{(1)}) \sim s$ and $\|M_\ell^{(1)}\|_0 = |\Theta_\ell|(1 - [\mathcal{Q}(s)]_\ell)$.

---

To circumvent this issue, we design an improved algorithm that produces embedded subnetworks on each iteration, allowing binary search to work (see Algorithm 1). Starting from the extreme subnetworks $S_1$ (fully-dense, corresponding to masks $\mathbf{M}^{(1)}$) and $S_2$ (fully-sparse, corresponding to masks $\mathbf{M}^{(2)}$), we narrow the sparsity gap between them while preserving $S_2 \subseteq S_1$ so that EffectiveSparsity$(S_1) \leq$ EffectiveSparsity$(S_2)$. For each layer, we keep track of unpruned connections $U_\ell$ of $S_1$ and pruned connections $P_\ell$ of $S_2$, randomly sample parameters $T_\ell$ from $U_\ell \cap P_\ell$ according to $\mathcal{Q}$ and form another network $S$ by pruning out $\bigcup_\ell T_\ell$ from $S_1$ (or, equivalently, reviving in $S_2$). Depending on where effective sparsity of $S$ lands relative to target $s$, we assign $S$ to either $S_1$ or $S_2$ and branch. Since connections to be pruned from $S_1$ (or revived in $S_2$) are chosen randomly at each step, weights within the same layer have equal probability of being pruned. Once $S_1$ and $S_2$ are only 1 parameter away from each other, the algorithm returns $S_1$, yielding a connected model.

