# OpenReview forum: "Connectivity Matters: Neural Network Pruning Through the Lens of Effective Sparsity"
_ICLR.cc/2022/Conference — ICLR 2022 Submitted_

### Official Review · Reviewer_Ne1A · 2021-11-01

**Correctness:** 3
**Technical Novelty And Significance:** 1
**Empirical Novelty And Significance:** 2
**Recommendation:** 3
**Confidence:** 4

**Main Review:**


Overview:

The issue of effectively disconnected parameters [Tenaka et. al.], while known, beckons the proper methodological accounting of parameters --- to those 'effectively' contributing. This paper describes this discrepancy and quantifies it at small scale.
While disconnections are known, and to this reviewer at least.--- known to be quantified by practitioners --- such quantification is indeed poorly documented and this paper eloquently captures it. Beyond the methodological importance of such explicit clarification, the practical relevance and expectation of their impact could use further discussion and motivation.

Major Pros:
1. The issue of proper methodology in the context of counting of parameters is important and by extension so is accounting for effectively defunct parameters due to disconnection.
2. The paper is well written and clear.

Major Cons:
1. This discrepancy (between effective and direct parameter count) is known (as the authors mention [e.g. Tenaka et al]) and thus the major contribution of this paper is of limited novelty.

2. While the actual methodological claim (one should account for disconnections via effective counting) is very much valid and orthogonal to experimental settings, the practical discrepancy (and thus implications) between direct/effective is very much dependent on the settings:

2.a. As mentioned by the authors (and demonstrated in figure 3), for the iterative pruning of Syntflow and SNIP-iter, no disconnection associated discrepancy appears. Tanaka et al single out the general role of the Iterative nature in upholding continuity even in methods which are not explicitly designed to preserve continuity.
In light of such an observation --- for SOTA, such as Iterative Magnitude Pruning (IMP) , it is not clear that in practice a large discrepancy would be present even up to higher than practically employed sparsity levels.

2.b. Further, in terms of architectures, probability of disconnection in modern architectures --- and especially those containing residual connections --- is much reduced. Indeed, in the Resnet experiments (figure 4) there seems to be a relatively small difference between effective and direct. As such, for practical architectures --- the level of actual discrepancy in practice is not demonstrated to be (or as expected to be) meaningful.

2.c. The paper demonstrated the results for small datasets (and weak networks) (mnist/lenet , VGG/cifar10-100, Resnet18/TinyImagenet).
The lack of practical large scale (Imagenet) and SOTA architectures limits the conclusions to the applicability.



**Summary Of The Paper:**

The paper highlights the difference between the explicit and effective (through disconnection) number of zeroed parameters in a model --- and argues it is the effective parameters that should be used as the benchmark for comparison.
Further comparing pruning methods and re-affirming their performance relative to random pruning equivalents. Finally introducing a layer allocation method which can serve as a better performing random baseline.


**Summary Of The Review:**

The paper eloquently presents and quantifies --- in the small scale --- the known issue of disconnections as affecting parameter count and pruning methods [Tenaka et al]. The paper can further engage with practical implications.
I agree that counting effective rather than direct parameters is methogologically good practice. However, the limited scale and demonstration  limits the insight applicability in practice --- especially, as SOTA methods and architectures have structural characteristics which call into question the the level of discrepancy (missing discussion/dimensions of exploration).

---

> ### Author Response · Authors · 2021-11-17
> **Author's Response**
>
> We thank the reviewer for their time and feedback. We address the issues raised by the reviewer below:
>
> 1. While effective sparsity is indeed a simple idea, we don’t agree that it is not novel. First, all (even the most recent) papers on network pruning report empirical results over direct sparsity, which, as we argue, disagrees with actual effective sparsity even in commonly used sparsity regimes and networks (see, e.g., Frankle et al., 2021). Second, even though SynFlow (Tanaka et al., 2020) does optimize for effective sparsity, it is seemingly a fortunate byproduct rather than an explicit intention. Indeed, Tanaka et al. aim to respect layer-wise conservation of synaptic saliency, which is not equivalent (though related to) effective sparsity. Furthermore, in Appendix 11 (page 16), they explicitly state that SynFlow produces parameter scores so that the corresponding per-layer sparsity avoids layer-collapse. This indicates that the authors are unaware of effective sparsity, for SynFlow’s competitive performance is due to the underlying structure of the network and can’t be explained by fortunate per-layer sparsities alone.
>
> 2. It is true that iterative pruning methods are generally less likely to develop inactive neurons. However, there are lots of types of pruning algorithms that are not iterative in nature, which is partly due to excessive costs associated with repeated pruning (e.g., IMP). The development of competitive and efficient pruning algorithms involves drawing comparisons between iterative and non-iterative approaches, and ensuring faithful benchmarking is essential. Effective sparsity is the only correct sparsity measure, while direct sparsity is fundamentally flawed. The discrepancy between the two, whether large or small, jeopardizes fidelity of research by potentially distorting experimental evidence. This is not desirable under any circumstance, especially given availability of effortless means of accounting for (computing and pruning with respect to) effective sparsity discussed in our work. The fact that recent publications on pruning report performance against incorrect sparsity measure (which, as we show, can be critically misleading) speaks for both novelty and importance of effective sparsity.
>
> 3. Finally, we acknowledge that it is important to consider both effective sparsity and IGQ in a variety of scenarios, including other network architectures and large-scale datasets. Since each new setup requires a large number of experiments (more than 100 training cycles per network per pruning algorithm), this study covers only some popular benchmarks and baselines from the relevant pruning literature (our experimental setup is very similar to that found in other papers, e.g., Su et al., 2020 and Tanaka et al., 2020, which do not report results with ImageNet). Experiments with other network types, application domains, and pruning strategies are left for future research. With this paper, we wish to initiate this research thread by clearly introducing the new concepts and providing an initial general experimental base.

---

> > ### Comment · Reviewer_Ne1A · 2021-11-20
> > **reply**
> >
> > I thank the authors for the discussion and reply, part of my comments are addressed though I remain unconvinced and others are not adequately addressed.
> >
> > With respect to novelty. I respectfully do not agree that it is coincidental ("seemingly a fortunate byproduct ...") that equating max and critical compression Tenaka et al [1] goes hand in hand with mitigating direct-effective disparity. It is a result, exemplified in the extreme scenario of disconnections. Beyond saliency --- further, the iterative nature effect has been recognized as pivotal to deal with these phenomena both in Tanaka et al [1] and in Verdenius et al [2]. In the authors' paper both synflow and SNIP-it correspondingly do not suffer disparity.
> >
> > So, not only is the phenomena recognized, key aspects for its mitigation have been algorithmically pointed out.
> >
> > Importantly, it is an insufficient condition for superior performance, as syntflow for example --- which the authors mention for its "SynFlow’s competitive performance" --- is far inferior to IMP.
> >
> > As to the cited works of Frankle et al, in these cases IMP likely does not result in disparities --- and in any case the authors do not show otherwise in the same settings.
> > The authors do not adequately address this: to iterate -- as practical networks (apropos with residual connections) , with methods where it is known explicitly how to address ailments associated with the issue of pruning important versus unimportant (indeed hurting saliency and at the extreme causing premature disconnections), are used for driving SOTA.
> >
> > The demonstration on weak methods/networks is not a matter of resources but rather is a potentially partial portrayal of the degree of already-mitigated aspects of the discussed phenomena in scenarios of interest. In other words Tenaka et al presented the problem at the small scale and demonstrated how it is --- in a principled novel way --- addressed. The small scale suffices for a constructive demonstration.
> > This paper implicitly implies that there is practical importance --- but leaves the question of 'is this a problem in practice?' open (even if one ignores the already existing principled mitigations). In the response the authors argue that this is an important problem because iterative methods are expensive for example. What is the role of architecture then? Apropos the suspicion that residual connections (practically ubiquitous in modern architectures) delay the disparity to practically non-interesting compression levels.
> >
> > I will stress again that I agree that acknowledging this phenomena is important, and the methodology and terminology the authors argue for are correct and important in my view. Indeed, works done to address sources of this phenomena have already been done as mentioned.
> >
> > [1] Pruning neural networks without any data by iteratively conserving synaptic flow
> > https://arxiv.org/pdf/2006.05467.pdf
> > [2] Pruning via Iterative Ranking of Sensitivity Statistics
> > https://arxiv.org/pdf/2006.00896.pdf

---

> > > ### Comment · Reviewer_B6nv · 2021-11-29
> > > **Thanks for your replies**
> > >
> > > I'd like to thank the authors for their replies. I've read these carefully but I still think the motivation, method, and the experiments in this paper should be improved before I can recommend accept. However, I will keep an open mind when discussing the paper with other reviewers and the ACs.
> > >
> > > Some points re your replies:
> > >
> > > * I think it's absolutely fine to include LeNet results given than SNIP used them as a baseline but I remain unconvinced that the VGG and ResNet experiments in this paper provide strong evidence for the claims.
> > > * The point you make about ensembling extremely-pruned model with poor performance is interesting but I haven't seen any evidence that this in fact works.
> > > * Thanks for clarifying the analogy from thermodynamics. I think the paper can be improved if you include this explanation in the paper. Although, I'm still not sure how F is not equal across layers.

---

### Official Review · Reviewer_xGEE · 2021-11-02

**Correctness:** 4
**Technical Novelty And Significance:** 2
**Empirical Novelty And Significance:** 2
**Recommendation:** 3
**Confidence:** 4

**Main Review:**

Strengths:
* The paper is clear and presents an interesting analysis about the discrepancy between pruning models and the resulting, underlying graph.
* The detail and analysis is thorough, and the methodology is clear and concise.
* The presented layer-wise sparsity quota Ideal Gas Quotas (IGQ) are interesting and produce compression distributions very similar to other methods (such as in Fig. 9).
Weaknesses:
* Many important points/figures in the paper are on very small datasets where models are extremely over-paramaterised. For example, Fig. 2 on MNIST would be much stronger, I think, on a slightly larger dataset (CIFAR-10, perhaps?)
* Overall, I found the results largely focused on models that do not reach competitive performance on larger datasets. ResNet18 on TinyImageNet is shown, but it'd be nice if more modern architectures were shown.

Comments:
* Would it be possible to adjust for effective FLOPs in a correctly pruned graph?
* It would be cool to investigate various choices and their impact on sparsity/effective sparsity during pruning. For example, I imagine models with residual connections will have slightly different dynamics.
* I'd be interested in a bit of a comparison of different model architectures-- for example, sparsity in convolutional filters will have a different impact in the effective sparsity metric than sparsity in MLPs (if I understand correctly). Therefore, there will be some differences between architectures that allocate parameters differently (and what is pruned).
* Showing how some of these methods look in another domain such as NLP could be interesting.

Minor comments:
* I found the caption to Figure 3 slightly confusing.

**Summary Of The Paper:**

The paper points out that in pruning, there is often a difference between the number of zero parameters and the number effective zero parameters due to disconnection during pruning. A thorough analysis is done comparing many pruning methods across MNIST, Cifar-10, and TinyImageNet using various architectures. The authors introduce a new physics inspired layerwise sparsity quota  baseline the "Ideal Gas quota" which sets a sparsity per layer.

**Summary Of The Review:**

The paper is well written, very clear, and well presented. The authors raise and clearly document the discrepancy between raw sparsity and the actual effective sparsity of a model. An interesting and compelling new baseline is presented.
However, the results are shown on small datasets without comparison with many modern architectures.
Replacing the MNIST/LeNet/VGG results with more modern architectures would clearly increase the impact of the paper.

---

> ### Author Response · Authors · 2021-11-17
> **Authors' Response**
>
> We thank the reviewer for their feedback and comments.
>
> 1. While it is true that we did not conduct experiments with large-scale datasets (e.g., ImageNet), we would like to emphasize that we followed our key references when choosing network architectures and datasets for experiments. For example, LeNet-300-100 on MNIST and LeNet-5 on CIFAR-10 appear in Lee et al. (2018). We present the full list of sources of our architecture-dataset combinations and hyperparameters in Appendix A.
>
> 2. We agree that experiments with other architectures and datasets would improve the paper. However, since each new setup requires a large number of experiments (more than 100 training cycles per network per pruning algorithm), this study covers only some popular benchmarks and baselines from the relevant pruning literature (our experimental setup is very similar to that found in other papers, e.g., Su et al., 2020 and Tanaka et al., 2020, which also don't include results on ImageNet). Therefore, experiments with other network types, application domains, and pruning strategies are left for future research. With this paper, we wish to initiate this research thread by clearly introducing the new concepts and providing an initial general experimental base.
>
> 3. Yes, it should be possible; thank you for the suggestion.
>
> 4. We agree that experimenting with other network types and layers could be a valuable contribution. In the current version of the paper, we already analyzed most general scenarios seen in related works: fully-connected, convolutional, and residual networks.
>
> 5. It is true that convolutional layers are less prone to inactive neurons than fully-connected ones. Although we provide no rigorous proof of this, we present intuition and empirical evidence (Figures 3 and 4) supporting the claim.
>
> 6. We agree that other domains such as NLP are interesting to investigate, but this is out of scope of this paper. Most recent works in pruning choose computer vision for benchmarking, and we follow the same strategy.

---

> > ### Comment · Reviewer_xGEE · 2021-11-20
> > **Thanks for your reply**
> >
> > I understand that many previous works have only shown results on fairly small scale datasets, though I think that this is a problem and overall this needs to change. I would really love to see a future where more modern architectures and datasets are experimented with, because the regime of some very over-parameterised networks is certainly different from what is more commonly used now.
> >
> > I also think that comparisons w.r.t. FLOPs would make things more interpretable and useful. For the time being, I am leaving my score unchanged.
> >
> > This is very exciting work, and I think with a bit of tweaking it will be a great paper.

---

### Official Review · Reviewer_B4nR · 2021-11-02

**Correctness:** 3
**Technical Novelty And Significance:** 3
**Empirical Novelty And Significance:** 3
**Recommendation:** 6
**Confidence:** 4

**Main Review:**

**Strong points:**

- From my experience, I recognize this problem of “effective sparsity” and, up to my best knowledge, this is the first time when I see it so well formulated. Thus, the paper was a pleasure to read, providing useful insights in understanding sparse neural networks and pruning techniques.

- Few small innovations are introduced to cope with the concept of “effective sparsity” in order to improve the trade-off model size/accuracy

- The code is provided for easy reproducibility

**Weak points and suggestions for improvement:**

- The paper contributions are hard to derive from the actual text. I suggest adding somewhere in the Introduction section, few short and concise bullet points to present the paper contributions and to sharply discuss what is novel with respect to the literature.

- The Related Work section contains just a categorization of various pruning techniques. I suggest to add an extra paragraph which clearly discuss the related work with respect to the “effective sparsity” concept.

- The experiments are performed just on dense-to-sparse training methods. I suggest considering also sparse-to-sparse training methods with prune and grow strategies. I agree with the authors statement from the Discussion section that it would be more difficult to incorporate their proposed effective pruning strategy in sparse training. This incorporation can be indeed let for future work, but just studying effective sparsity in sparse training should not be so difficult and the results can be of interest (e.g., same as in Figure 3).

- I understand the ERK baseline for the proposed IGQ method but decoupling ERK from sparse training (as it was originally designed) may lead to misleading results interpretation. Nothing to change in the experimental section, but I suggest clarifying (discussing better) this aspect.

- It is not very clear to me if the paper suggests that the connectivity pattern itself is also quite important in ensuring a good trade-off model size/ performance and not just the sparsity distribution. A discussion has been started on this topic in the last paragraph of page 2, but I would expect the Discussion section to come back to this topic and perhaps some empirical validation would be necessarily to support better the statement “…but find the truth to be more nuanced at higher compression rates…”. I wouldn’t expect a detailed study as it is outside of the scope of this paper, but rather some hints in order to avoid cutting from start future works on this topic.


**Summary Of The Paper:**

This paper raises a warning signal on an aspect which is typically ignored in sparse neural networks, i.e., the “unconnected” connections (the ones which do not belong to the paths connecting the input to the output neurons). Consequently, the paper studies the so-called “effective sparsity”, to formally understand the efficiency of various pruning methods to obtain sparse neural networks while considering the connections which contribute to the neural network inference (the ones which do belong to the paths connecting the input to the output).  Complementary, the paper proposes some (minor) contributions to cope with effective sparsity. Overall, the paper is well written and has well-designed experiments.



**Summary Of The Review:**

Overall, I believe that this is a well-written paper, with a clear message, which also has some space for improvement.

---

> ### Author Response · Authors · 2021-11-18
> **Authors' Response**
>
> We thank the reviewer for their thorough evaluation and the time spent reading our paper.
>
> 1. Each paragraph in Section 1 that starts with boldface describes one contribution of the paper. We added an explicit summary at the end of Section 1 in the revised version.
>
> 2. To our knowledge, we are the first to formulate the concept of effective sparsity, and we have not seen any works addressing this matter. While some studies (e.g., Tanaka et al., 2020 and Jorge et al., 2021) present pruning methods that fortuitously avoid inactive neurons, they do not discuss effective sparsity as a factor in their analysis. Could you please list the papers you have in mind?
>
> 3. We agree that benchmarking more types of pruning methods is interesting and necessary, but we let it for future research. We would also like to note that most methods we considered are in fact sparse-to-sparse (i.e., prune before train), but we agree that algorithms that learn a subnetwork’s topology along with weights are interesting to consider.
>
> 4. We agree that ERK was originally introduced as layerwise sparsity quotas for algorithms that optimize structure and parameters concurrently (i.e., SET by Mocanu et al., 2018 and RigL by Evci et al., 2020). However, ERK has already been used as a stand-alone sparsity distribution method in previous works (e.g., Lee et al., 2021). We added a note about this to the updated version.
>
> 5. The sentence from Section 1 you are referring to is meant to provide a one-sentence summary of Figure 5: the connectivity structure within layers of subnetworks produced by SNIP, GraSP and SynFlow can be reshuffled without sacrificing trained performance up to 0.99 sparsity. At higher sparsities, however, test performance does sometimes become worse when random pruning is used (e.g., SNIP on LeNet-300-100 and SynFlow on VGG-19). This degradation is not at all surprising in the case of SynFlow because random pruning within layers (or, equivalently, layerwise reshuffling) no longer guarantees integrity of the network and may introduce inactive connections that SynFlow provably avoids.

---

> > ### Comment · Reviewer_B4nR · 2021-11-29
> > **Thanks for answering**
> >
> > I thank the authors for answering to my concerns and for clarifying the novel contributions of this paper. Indeed, I was thinking that this paper is the first one which formulates clearly the concept of "effective sparsity", but I wasn't completely sure. At the same time, even if the paper in its current state is quite informative, I believe that it can still be improved on some aspects according with some of the discussions from this peer-review process. Consequently, I will keep my initial score "6 - marginally above the acceptance threshold". It would be nice to see this paper accepted, but unfortunately I cannot increase my recommendation to 8 as I believe that the paper is not yet at that level.

---

### Official Review · Reviewer_B6nv · 2021-11-04

**Correctness:** 3
**Technical Novelty And Significance:** 1
**Empirical Novelty And Significance:** 1
**Recommendation:** 3
**Confidence:** 4

**Main Review:**

While indirect pruning is a reasonably known phenomenon, it is an interesting question to see whether it can give us any insights when comparing various pruning methods. Unfortunately, this paper doesn't do a thorough job doing that due to choice of baselines and poor motivation of extreme pruning regime:

- I am surprised to see LeNet-300-100 as baselines here. I do not want to be the "show me imagenet results" reviewer, but I feel the choice of MNIST, and LeNet architectures are not justified here. Indirect pruning happens in extreme pruning regimes and I am assuming we care about that for very, very large networks. LeNet-300-100 is tiny and does not have convolutions, so is this a good baseline to draw conclusions from?
- In the introduction, the authors do say that typical compression rates are 10x (0.9 pruning) and 100x (0.99 pruning) and motivate extreme pruning because of models with billions of parameters, yet their biggest models are ResNet-18 and VGG-19, where indirect pruning does not seem to be as big of a problem anyway.
- Overall, the authors should do a better job motivating extreme pruning regimes. Pruning is interesting because we think models are over-parameterized, and we can get "comparable performance" while pruning away a good portion of parameters. However, the performance in this extreme regime is very poor, and the architectures are simply "trainable". So why is this an interesting problem to look at?

**Structured pruning and CNNs**
- I think the authors should also clarify the link between indirect pruning and structured pruning. Isn't the phenomena described by the authors the intended behaviour in structured pruning? It would be great if authors could also compare with structured pruning methods but I think at least the link between the two should be discussed in the paper.
- The authors do not discuss what happens to the disconnected neurons as a result of indirect pruning. Can you set their value to zero (i.e. actually prune them)? Can their value always be fused into the next layer? or perhaps into the batch norm parameters?
- The authors also don't discuss convolutional networks in details. I'm assuming indirect pruning is less of a problem there because you'd have to prune an entire channel to have indirect pruning. Given that most of our commonly used architectures and other pruning baselines use CNNs it seems crucial to have it analysed in detail and not rely on MLPs to make conclusions

**Layerwise Sparsity Quotas (LSQ) and allocation method IGQ**

I don't understand the Ideal Gas Law approach at all! What is the motivation for borrowing this from thermodynamics? What is the justification that neural networks behave like gasses? It is one thing to mention this in passing as an inspiration source, but when you go as far as mapping a neural network to a system in thermodynamic equilibrium, the height of a cylinder to the number of weights, and pruning to applying an external force to then you should provide better justifications. Moreover, the final derivation for prune ratio of each layer is $F | \Theta _l | + 1$. There are two things I don't understand about this: (1) How is $F$ related to the desired pruning target? The authors say $F$ can be found via a binary search from pruning target but do not elaborate. (2) No matter how F is found I'm assuming it's a global value and not a per-layer one. Doesn't this reduce the formula to simply applying pruning uniformly??

**Other things**
- The authors use VGG19. I have seen different variants of VGG-19 for non-imagenet datasets. In Some variants there's a single dense layer after convolutional layers, and in some other variants there are multiple dense layers. Could the authors confirm which architecture they've used? This makes a big difference in prunability of the network and I'm curious to know if the indirect pruning is mostly happening in the final dense layers or in convolutions as well.
- In the discussion section the authors argue that "effective compression" is the "correct" measure for pruning algorithm. I think that's a strong claim that's not well supported in the paper. You should at least clarify that it's aimed at extreme pruning regime (and probably non CNN based architectures?
- The caption in Figure 2 says "SynFlow has a better sparsity-accuracy tradeoff than SNIP" Isn't the opposite true? I guess it's a typo
- In figure 3, would things look too weird if you use the same units in the x axis and y axis so that we can compare the effect across architectures more easily?
- In figure 5 could you rename "Random" to "SNIP + RandomReshuffle" or something. I think random is misleading here given that you're still applying layer-wise pruning ratios.

**Summary Of The Paper:**

This paper highlights the fact that when doing unstructured pruning with extreme sparsity targets, we end up pruning entire neurons, which effectively detaches neurons from the input/output. The authors argue that the effective sparsity levels in these cases is higher than our target levels, and taking it into accounts exhibits differences between various pruning method that are otherwise not obvious.

**Summary Of The Review:**

This paper draws too many conclusions based on very small networks, datasets, and for a pruning regime where the network is simply trainable and has nowhere near the performance of the unpruned model. Overall I'm not convinced that "effective pruning", and extreme pruning gives us a enough insight to justify accepting the paper.

---

> ### Author Response · Authors · 2021-11-16
> **Authors' Response (Part 1/2)**
>
> We thank the reviewer for valuable comments and would like to address their questions and concerns:
>
> 1. Our choice of network architectures and datasets was mainly inspired by previous works. In particular, LeNet-300-100 and LeNet-5 appear in the experimental section of SNIP (Lee et al., 2018), which is one of the methods we review. Moreover, as hypothesized by the reviewer and confirmed in our study, the gap between effective and direct sparsities largely depends on the architecture type. Thus, we believe our paper would have been incomplete without experiments on LeNet-300-100, which serves as a representative of fully-connected networks. Finally, we would like to emphasize that our analysis goes well beyond LeNet architectures and includes more recent networks such as ResNets and VGGs.
>
> 2. We agree that, on average, effective sparsity is most critical in extreme compression regimes (x100 and higher), which become more practical in larger networks. However, the goal of our work is to introduce the long overlooked concept of effective sparsity and initiate reevaluation of different pruning methods applied to different architectures in light of this new metric. That is, the experimental body of our paper is by no means exhaustive and is meant to lay the foundation for future research. Further, we would like to note that Tanaka et al. (NeurIPS 2020) study extreme compression regimes by using similar-sized models (ResNet-18, WRN-18, VGG-11, and VGG-16). Therefore, our experimental base is in line with current research in this domain. Finally, as we argue below, the role of effective sparsity depends more on architecture type and not necessarily on model size.
>
> 3. We agree that extremely sparse networks may suffer unreasonable performance degradation. However, as we note in Section 6, such lightweight but not so well performing models could be beneficial in ensembling. Most importantly, our paper follows an emerging trend on extreme pruning (Tanaka et al., 2020; Jorge et al., 2021). Thus, even though it has not yet found its practical applications, extreme pruning is certainly interesting to the research community.
>
> 4. Effective sparsity is exactly equal to direct sparsity in the context of structured pruning, which we briefly mention is Section 2. Therefore, we think that there is limited value in further discussing structured pruning in our work. On the other hand, we agree that comparing structured and unstructured pruning methods is interesting in and of itself, however, this task is not in the scope of our paper.
>
> 5. We disagree that our paper does not address the problem of inactive neurons and ways to deal with them. In Section 3, we discuss reasons for why inactive neurons emerge under some pruning methods (SNIP, GraSP, etc.) and not under others (iterative-SNIP, SynFlow). In particular, we state that connections attached to inactive neurons receive the lowest possible score of 0 by certain pruning criteria and that they are automatically set for removal when pruning is performed iteratively. We would like to emphasize this has not been known yet as Tanaka et al. (2020) did not attribute the exceptional performance of  their algorithm (SynFlow) in extreme sparsities to this phenomenon. Finally, we describe the algorithm (code available in supplemental materials) that efficiently finds inactive neurons and prunes them from the network.
>
> 6. We disagree that the phenomenon of inactive neurons in convolutional networks is not addressed in the paper. Indeed, we discuss in Section 3 that, as the reviewer correctly suggests, it is more difficult to disconnect feature maps (compared to neurons in fully-connected layers) from the computational flow as they are “protected” by more parameters (i.e., all parameters in all kernels have to be removed to inactivate a feature map, which is likely to happen only at large sparsities). We would like to assure the reviewer that our conclusions are not based solely on LeNet-300-100, which is the only fully-connected network among the other four convolutional architectures used in our study.

---

> > ### Author Response · Authors · 2021-11-16
> > **Authors' Response (Part 2/2)**
> >
> > 7. We would like to clarify the reviewer’s questions with regards to Ideal Gas Quotas and Section 4 at large. The analogy from thermodynamics is indeed merely an intuition and does not bear strong fundamental connections between ideal gases and neural networks. Nevertheless, the motivation to use this model comes from certain reasonable requirements  that a proper layerwise sparsity distribution must meet. These requirements are listed in Section 4 and are captured in Definition 1. Moreover, we required that our layerwise sparsity quotas assign higher sparsity to larger layers, which comes from observing layerwise compression distributions of successful pruning methods (SynFlow and LAMP) in Figure 9. All these basic requirements are clearly satisfied by the Ideal Gas Model. The detailed derivations presented in Section 4 are given for sake of completeness and as an opportunity to build more intuition about our method. We now answer specific questions from the reviewer: (1) As can be seen from our derivations, $F$ controls compression of each individual layer, and, in particular, the overall network compression. Moreover, it is clear that smaller values of $F$ result in lower compression and larger values of $F$ result in larger compression. Hence, we can start with some small ($F=0$, which corresponds to unpruned network) and some large (e.g., $F=10^{30}$, which corresponds to fully pruned network) values of force and quickly converge to the one that yields the desired level of overall compression by using binary search. (2) Force $F$ is indeed a global value, as discussed above. Once the appropriate value of $F$ is identified, one finds the corresponding per-layer compressions ($1+F|\Theta_{\ell}|$), which are not necessarily equal across layers.
> >
> > 8. Following Wang et al. (2020), our version of VGG-19 does not have fully-connected layers following convolutional ones except for the output layer. We state sources of our architectures and hyperparameters in Appendix A.
> >
> > 9. We strongly disagree that effective sparsity as the correct sparsity measure applies only in extreme compressions or to non-convolutional architectures. As we said in part 2 above, effective sparsity is indeed more impactful in these scenarios, however, it is still correct and faithful while direct sparsity is not. The discrepancy between direct and effective sparsity, however small or large, varies from case to case, but the latter is always correct and hence should be used by researchers and practitioners.
> >
> > 10. The caption of Figure 2 states that SynFlow has a better accuracy-sparsity trade-off compared to SNIP *when plotted against direct compression (dashed curves) but not when plotted against effective compression (solid curves)*. There is no typo in the caption.
> >
> > 11. Could you please specify what units you would want the axes to be? Currently, they both represent compression (direct on x-axis and effective on y-axis), so that units are in fact the same.
> >
> > 12. The pruning procedure used in experiments displayed in Figure 5 does involve random pruning while respecting specified layerwise sparsity quotas. Note that this is equivalent to layerwise reshuffling, which we believe is a less standard term than simply random pruning.
> >
> > We hope that our responses address concerns of the reviewer, and they will consider increasing their score.

---

### Decision · Program_Chairs · 2022-01-20

**Decision:**

Reject

**Comment:**

### Summary

This work investigates effective sparsity: an assessment of the sparsity of pruned networks that accounts for the fact that unpruned neurons can still be completely disconnected through pruning.  Hence, the effective sparsity of a network may be much lower than otherwise reported.

### Discussion

#### Strengths

- The paper studies an important metric that deserves additional attention in the community, where a change in metric may guide either the theory or practice of pruning.

- The paper evaluates direct versus empirical sparsity for a healthy number of pruning techniques.


#### Weaknesess

- While this paper appears to be the most direct study of effective sparsity at the moment, it is not the first. Appendix M of [1] defines effective sparsity and shows that direct and effective sparsity are similar for contemporary pruning at initialization techniques. However, that work does not evaluate random pruning. This work here will need to revise its novelty claims to account for these results as its characterization that [1] only considers direct sparsity is incorrect.


- "Computing effective sparsity:" the procedure in question is similar to that of Appendix M in [1], thus its relationship should be detailed.


- With the primary observations residing in the regime of extremely sparse neural networks, the elements of the response (and in the last paragraph of the paper) that claim this regime is productive for ensembling should make a more prominent appearance in the introduction of the work.

[1] Pruning Neural Networks at Initialization: Why Are We Missing the Mark?
Jonathan Frankle, Gintare Karolina Dziugaite, Daniel Roy, Michael Carbin. ICLR, 21


### Recommendation

I recommend Reject. Generally, the paper is well-written and the empirical characterization of direct versus effective sparsity is thorough (except for ResNet-50 results). However, the results and the language around these results need significant rescoping to account for novelty and the relation of the work to an area in which it is anticipated that these results will change theory, practice, or thinking (e.g., ensembling).

Though I cannot speak for future reviewers, IMO, an extension of the results here to ResNet-50+ImageNet should suffice to establish the extent of the discrepancy between direct and effective sparsity. However, to satisfy additional demands from reviewers for more practical relevance, I suggest an evaluation that demonstrates a consequential difference in behavior for a task that maps more closely to the anticipated area of impact (e.g., ensembling)